# PeaT1 and PeBC1 Microbial Protein Elicitors Enhanced Resistance against *Myzus persicae* Sulzer in Chili *Capsicum annum* L.

**DOI:** 10.3390/microorganisms9112197

**Published:** 2021-10-21

**Authors:** Khadija Javed, Talha Humayun, Ayesha Humayun, Yong Wang, Humayun Javed

**Affiliations:** 1Department of Plant Pathology, Agriculture College, Guizhou University, Guiyang 550025, China; khadijajaved829@gmail.com; 2Department of Environmental Science, PMAS-Arid Agriculture University, Rawalpindi 46000, Pakistan; 3Department of Surgery (Surgical Unit 1 HFH), Rawalpindi Medical University, Rawalpindi 46000, Pakistan; talhahumayun7@gmail.com; 4Department of Clinical studies, Pir Mehr Ali Shah-Arid Agriculture University, Rawalpindi 46300, Pakistan; ayeshahumayun221@yahoo.com; 5Department of Entomology, PMAS-Arid Agriculture University, Rawalpindi 46000, Pakistan; hjhumayun@gmail.com; 6Rothamsted Research, West Common, Harpenden AL5 2JQ, UK

**Keywords:** PeaT1, PeBC1, *M. persicae*, induced plant resistance against aphids, defense pathways, EPG

## Abstract

The green peach aphid (*Myzus persicae* Sulzer), a major and harmful chili aphid usually managed using chemical pesticides, is responsible for massive annual agricultural losses. The efficacy of two protein elicitors, PeaT1 and PeBC1, to stimulate a defensive response against *M. persicae* in chili was studied in this study. When compared to positive (water) and negative (buffer, 50 mM Tris-HCl, pH 8.0) controls, the rates of population growth (intrinsic rate of increase) of *M. persicae* (second and third generations) were lower with PeaT1- and PeBC1-treated chilli seedlings. *M. persicae* demonstrated a preference for colonizing control (12.18 ± 0.06) plants over PeaT1- (7.60 ± 0.11) and PeBC1 (6.82 ± 0.09) treated chilli seedlings in a host selection assay. Moreover, PeaT1- and PeBC1-treated chilli seedlings, the nymphal development period of the *M. persicae* was extended. Similarly, fecundity was lowered in the PeaT1- and PeBC1-treated chilli seedlings, with fewer offspring produced compared to the positive (water) and negative controls (50 mM Tris-HCl, pH 8.0). The trichomes and wax production on the PeaT1 and PeBC1-treated chilli leaves created a disadvantageous surface environment for *M. persicae*. Compared to control (30.17 ± 0.16 mm^−2^), PeaT1 (56.23 ± 0.42 mm^−2^) and PeBC1 (52.14 ± 0.34 mm^−2^) had more trichomes. The levels of jasmonic acid (JA), salicylic acid (SA), and ethylene (ET) were significantly higher in the PeaT1- and PeBC1-treated chili seedlings, indicating considerable accumulation. PeaT1 and PeBC1 significantly affected the height of the chili plant and the surface structure of the leaves, reducing *M. persicae* reproduction and preventing colonization, according to the data. The activation of pathways was also part of the defensive response (JA, SA, and ET). This present research findings established an evidence of biocontrol for the utilization of PeaT1 and PeBC1 in the defence of chili plants against *M. persicae*.

## 1. Introduction

Herbivores and plants have developed a complex interaction over the course of their respective evolutions. Plants are damaged by herbivores and thus modify their physical structures to cope with this damage; as such, plants have developed an economical defense mechanism to protect themselves against herbivores [1]. Herbivores’ development, colonization, feeding, survival, and oviposition are all affected by the structures and substances, which also attract natural adversaries and urge them to develop defence mechanisms [1,2]. Plants have established two primarily constitutive defensive mechanisms to deal successfully with this harm [1]. Plants use physically compromised barriers to resist colonization, such as cuticle trichomes, callose, cell walls, and suberin, but anti-biotic allelochemicals impact or stimulate pest production, fertility, and insect durability [3].

Aphids are phloem-feeding insects that spread plant viruses by syphoning off plant sap, resulting in major agricultural losses [4,5]. Aphid defence responses have been explored in a variety of aphid–plant settings. In green peach aphids *M. persicae*, *Arabidopsis thaliana* was shown to be less viable. Infested leaves with Sulzer [6]. Dietary effects were generated in chilli plants, and volatile organic compounds were released, resulting in a repellent effect against infested *Bemisia tabaci* [7]. *Brevicoryne brassicae* resistance reduced the survival rate and population growth parameters of immature *Plutella xylostella* in *Brassica napus* [8].

Jasmonic acid (JA), salicylic acid (SA), and ethylene stimulate the defensive response in plants (ET) [9]. In plants, salicylic acid (SA) and jasmonic acid (JA) are essential regulators of the induced defensive response [9,10]. The defense against sucking–piercing insects has been associated to SA, while the defense against chewing insects has been linked to JA [11]. ET regulates a variety of defense-related mechanisms in plants [12]. *Danaus plexippus* promotes JA pathway activation but inhibits SA acquisition in oleander aphids, *Aphis nerii*; JA has the opposite effect in *Asclepias syriaca* [12]. Plant responses to herbivory and necrotrophic disease infestations, according to current knowledge, activate the JA and SA defence pathways [10]. Similarly, some elicitors and eliciting components in plants can behave as resistant protein- and nucleotide-binding factors, resulting in aphid resistance [13]. Only a few prior studies have shown that JA and SA have a role in aphid response induction via increased expression of genes including *PR-1, PR-2, CHIT1, LOX1,* and *PAL*, all of which have been identified as responses induced by JA–SA after aphid feeding [14,15].

*M. persicae* Sulzer, a major harmful pest of cucumber, maize, barley, wheat, and beans in China, has a direct impact on crop productivity and quality due to its feeding behaviour. Plant defence responses are triggered by biotic and abiotic elicitors [16]. Elicitors are linked to a variety of diseases, including fungus, bacteria, viruses, and oomycetes. The most prevalent elicitors are proteins, glycoproteins, peptides, lipids, and oligosaccharides [17]. They are divided into two categories: race-specific groups that only elicit a defense response in host plants, and general defence groups that stimulate a defense response in both host and non-host plants [18]. Elicitors are bio factors or chemicals that plants use as signal molecules to promote systemic acquired resistance to diseases or herbivores by activating multiple defensive pathways [19,20]. Microbe-associated molecular patterns (MAMPs) and herbivore-associated molecular patterns (HAMPs) created by herbivorous insect pests are both examples of elicitors. The majority of HAMPs have been identified in pests that are lepidopterous, dipterous, or orthopterous [19]. Volicitin, for example, was discovered in beet armyworms (*Spodoptera exigua*) as the first herbivore-induced elicitor [21]. Elicitor proteins (MAMPs) from fungal (e.g., Pep-13 and endo-1,4-xylanases from *Phytophthora* and *Trichoderma*, respectively) and bacterial pathogens (e.g., flg22 from bacterial flagella) diseases have also been discovered [22,23]. These elicitors play an important role in crop protection because they can induce pest resistance, reduce pest fitness, and limit pest feeding. Elicitors are proteins, glycoproteins, and lipoproteins that activate signalling pathways, the hypersensitive response (HR), and reactive oxygen and nitrogen species (ROS and RNS) responses in plants to generate resistance to diseases and herbivore pests [20,24,25]. Reactive oxygen species (ROS) and nitric oxide (NO), both of which govern metabolic and transcriptional changes, are produced by physiological responses to common processes such as protein phosphorylation or plasma membrane protein activation [20]. Because of the rising demand for food safety, numerous protein elicitors have been investigated as potential pesticide substitutes [26,27,28,29].

PeaT1, broadly specific elicitor examined in *Alternaria tenuissima*; PeBC1 in *Botrytis cinerea* and is thought to promote plant resistance via the JA and SA pathways. It activates defense enzymes and strengthens cell walls while also stimulating the production of other defense-related genes [30,31]. Because of their minimal mammalian toxicity and excellent host specificity, entomopathogenic fungi are crucial in the biological control of insect pests [32]. Furthermore, these fungi have the ability to develop as entophytes within various plant parts [32,33]. They also produce systemic resistance in plants against biotic stressors such as phytoparasites, diseases, and nematodes [34], Furthermore, entomopathogenic fungi boost plant development [35], upsurge in the yield [36], and increase the nourishment of plant [37] and the growth of roots [38,39]. Abiotic stresses such as drought [30], iron chlorosis [40], and salinity stress [41] are also mitigated by these fungi. Fungi’s ecological functions have the ability to boost plant health and provide a new perspective on developing novel plant protection techniques [40]. Similarly, certain elicitor proteins from necrotrophic and biotrophic fungal infections have recently been identified, exhibiting induced tolerance to pathogens and herbivores. For example, in *A. thaliana*, the elicitor PeBC1, which was cloned from the necrotrophic fungus *B. cinerea*, generated disease resistance [42,43]. PeaT1 (GenBank: EF030819.1) is a type of general elicitor isolated from *A. tenuissima*. It activates systemic acquired resistance via the SA pathway in plants, resulting in cell wall strengthening and the upregulation of defense-related genes and activation of defense enzymes [44,45]. PeaT1 has been shown to promote growth and strengthen resistance to abiotic stresses in wheat and rice plants [46]. The aim of this study was to look into the activity and molecular mechanism of the *B. cinerea*-derived elicitor protein PeBC1 and the *A. tenuissima*-derived elicitor protein PeaT1 in the induction of green peach aphid resistance in chili plants. The impacts of PeaT1 and PeBC1 on *M. persicae* control, as well as the roles and mechanisms of PeaT1 and PeBC1 on *M. persicae* control, are investigated in this work to analyze the prospective influence of PeaT1 and PeBC1 on *M. persicae*. Trichomes were discovered on the leaf’s surface structure, thus prompting researchers to examine the contents of the JA and SA gene expressions from JA and SA. This research also includes information on PeaT1 and PeBC1 function, mechanism, and effects in the integrated management of the green peach aphid (*M. persicae*).

## 2. Materials and Methods

### 2.1. Chili (Capsicum annuum L.) Plants and Aphid Preparation

*M. persicae*, commonly known as the green peach aphid, was collected from a cabbage field and transferred to chili seedlings (*C. annuum* L.) A colony of green peach aphids, *M. persicae*, was maintained on chili (*C. annuum* L., XianJiao 8819 cultivar) plants in isolation cages in a growth chamber with a 16:8 h light/dark photoperiod, 60 percent relative humidity (RH), and 23 ± 1 °C. To ensure that aphids were properly adapted to the chemistry of the chilli plants, the colony was developed for three months prior to the start of the tests. Chili seeds (*C. annuum* L.) were sterilized in 75% ethanol for 20–30 s and then washed with distilled water before being pre-soaked in distilled water for 2–3 days before use. 

### 2.2. Expression and Purification of PeaT1 and PeBC1

PeaT1 was produced with the recombinant vector pET28-His-PeaT1 (Novagen, Burlington, MA, USA) in *Escherichia coli* host strain BL21 (TransGen Biotech, Beijing, China). PeBC1 was produced with the recombinant vector pET-28a-c His-PebC1 (Novagen, Burlington, MA, USA) in *E. coli* host strain BL21 (TransGen Biotech, Beijing, China). The pellets were removed, and the supernatant cells were resuspended and sonicated using an ultrasonic disruptor. Supernatant was collected and filtered with filter paper (size 0.22 m) after the solution was centrifuged at 12,000 rpm for 15 min. The elicitor proteins PeaT1 and PeBC1 were purified using the Akta Explorer Protein Purification System (Amersham Biosciences, Temecula, CA, USA) and a His-Trap HP column (GE Healthcare, Waukesha, WI, USA) with various loading buffers (A–D) as described by Wang et al. [31]. Other elicitors were promptly washed off the column with buffer A (50 Mm Tris-HCl, 8.0 pH), and buffer B was employed to stabilize the column (50 Mm Tris-HCl, 200 Mm NaCl). We employed buffer C (50 Mm Tris-HCl, 200 Mm NaCl, and 20 Mm imid-azole, pH 8.0) and elusion buffer D (50 Mm Tris-HCl, 200 Mm NaCl, and 20 Mm imid-azole, pH 8.0) for elicitor protein elution (50 Mm Tris-HCl, 200 Mm NaCl, and 500 Mm imidazole, pH 8.0) Wang et al. [31] described desalting the PeaT1 and PeBC1 elicitor protein in a HiTrap desalting column (GE Healthcare, Waukesha, WI, USA). A 12 percent SDS-PAGE resolving gel was used to determine the molecular mass of the purified elicitor proteins PeaT1 and PeBC1, and a GenStar M223 protein marker (5–245 kDa) was utilized to estimate the molecular mass of the pure recombinant PeaT1 and PeBC1 elicitor proteins.

### 2.3. Bradford Assay for Determination PeaT1 and PeBC1 Protein Concentrations

A Bradford assay was used to check the concentrations of both elicitor protein quantities via a BCA kit and the proteins were then kept at −80 °C for future use. The elicitor proteins were diluted 15, 25, 50, and 100 times before being used. PeaT1 concentrations were found to be 80.65, 48.39, 24.19, and 12.09 μg mL^−1^, while PeBC1 concentrations were 70.12, 42.07, 21.03, and 10.51 μg mL^−1^.

### 2.4. The Population of M. persicae in PeaT1 and PeBC1

Chili seeds were soaked for 24 h in different concentrations of PeaT1 (80.65 μg mL^−1)^ and PeBC1 (70.12 μg mL^−1^) solutions, respectively. Six seeds in a single pot were grown in organic soil (Flora guard SUBSTRAT). Three-week-old seedlings with different concentrations of PeaT1, i.e., 80.65 μg mL^−1^ and PeBC1, i.e., 70.12 μg mL^−1^, solutions were sprayed and then inoculated with 10–12 adults of *M. persicae* per plant after 24 h. Seedlings were sprayed every 7 days. Following inoculation, the number of settled aphids was counted every 5 days for *M. persicae*, as stated by Li et al. [47]. Following inoculation, the number of settled aphids was counted every 5 days for *M. persicae*, as stated by Li et al. [47] Positive and negative controls were water and buffer (50 mM Tris-HCl, pH 8.0). The study employed a CRD randomized statistical design. Seedlings from each plant were separated using transparent air-permeable cages. The experiment was repeated twice with four replications each time, i.e., for PeaT1, a different experiment was repeated twice with four replications, and for PeBC1, a separate experiment was repeated twice with four replications. 

### 2.5. Feeding Behavior of Aphid M. persicae (Choice Test)

Chili seeds were soaked for 24 h in different concentrations of PeaT1 (80.65 μg mL^−1^) and PeBC1 (70.12 μg mL^−1^) solutions, respectively. In organic soil, three seeds were grown in a single pot (Flora guard SUBSTRAT). Three-week-old chili seedlings were sprayed with different doses of PeaT1 and PeBC1 solutions (80.65 μg mL^−1^ and 70.12 μg mL^−1^. The PeaT1-treated chilli seedlings and control seedlings were placed in a clear breathable cage (60 × 60 × 60 cm) with cross-touch leaves and a white cardboard bridge (12 × 4 cm) linking the base section of the stems. PeBC1-treated and control seedlings were placed separately in a transparent breathable cage (60 × 60 × 60 cm) with cross-touch leaves and a white cardboard bridge (12 × 4 cm) linking the stem base portion. The study employed a CRD randomized statistical design. For both experimental studies, independently thirty wingless *M. persicae* adults were released in the center of the bridge. The experiment was repeated 15 times, with the aphids on each seedling being counted after 24 h. 

### 2.6. The Intrinsic Rate of Increase in M. persicae by PeaT1 and PeBC1

Chili seeds were soaked in 80.65 μg mL^−1^ of a purified elicitor protein PeaT1 and 70.12 μg mL^−1^ of a purified elicitor protein PeBC1 solution for 24 h and then transferred for sprouting 2–3 days in distilled water in Petri plates. A CRD randomized statistical design was used. Seedlings were sprayed after 24 h with 80.65 μg mL^−1^ of the PeaT1- and 70.12 μg mL^−1^ of the PeBC1-purified protein solution. Inoculation with a freshly born *M. persicae* nymph was then carried out for every seedling. All seedlings were separated by a glass tube cotton gauze, and the movement of aphids was restricted on the leaf in a plastic ecological cage (2.7 × 2.7 × 2.7 cm). The edge of the ecological cage was covered with a sponge to avoid causing mechanical wounds to the leaf. The instar of the aphid was checked every 12 h to record the time when it produced the first nymphs. Then, the number of newborn nymphs was recorded twice a day to record the total time and number of offspring produced, which were removed every day after each count to avoid crowding. After 5 days, the same test was conducted on seeds and seedlings. With 30 replicates per treatment, the experiment was repeated twice individually. The increase in each aphid’s intrinsic rate was measured using the following formula:*r_m_* = 0.738 × (ln *Md)/Td*

*Md* is the number of newborn nymphs in a development time equal to *Td*, which is the time between the birth of an aphid and its first reproduction.

### 2.7. Investigation of Feeding Behavior of M. persicae by (EPG)

The chili seeds were submerged and germinated in distilled water for 3–4 days, as previously described. Similarly, until day 7, sized seedlings were individually put into organic soil. An electrical penetration graph (EPG; GIGA-8d) was then employed on wingless, 12–15-day-old, healthy adult *M. persicae* 24 h after spraying the seedlings. All aphids were starved for one hour before to the test. The studies took place every day for 4 h at the same time. The EPG was performed from 10:00 to 14:00 every day and recorded continuously for 4 h. All experiments were carried out in a Faraday cage at 20 ± 1 °C. The visualization and manual labeling of the various aphid-feeding waves were determined and manually studied using an A B stylet. A wave identifier was also used, as previously described [48].

### 2.8. M. persicae Bioassay by PeaT1 and PeBC1

A bioassay of the PeaT1 and PeBC1 elicitors against *M. persicae* on chili plants was carried out with different concentrations of the protein-purified solution, namely 80.65, 48.39, 24.19, and 12.09 μg mL^−1^ and 70.12, 42.07, 21.03, and 10.51 μg mL^−1^, respectively. Different protein concentrations were determined using the Bradford assay. Approximately 2–3 mL of PeaT1 and PeBC1 were applied to the chili plants at the three-leaf stage with a separate spray bottle until the solution was drained off the plants. Waters and buffers (50 mM Tris-HCl, pH 8.0) were used as positive and negative controls. 3–5 freshly moulted 0–6 h aphids were allowed to feed on the plants after they had dried overnight. The total number of offspring produced by all aphid instars was calculated as the total number of days in which aphids lived, and the time of nymph development was observed by consecutive observations at intervals of 3 h until the bioassays were completed for each instar as the total number of offspring produced by all aphid instars. A factorial ANOVA with least significant difference (LSD) at α = 0.05 was used to compare the data statistically. With 10 replicates per treatment, bioassays were repeated independently at three non-identical temperature regimes (23, 25, and 27 °C).

### 2.9. Effect of PeaT1 and PeBC1 on Physical Structure of Chili

Chili seeds were soaked for eight days in different concentrations of PeaT1 (80.65 μg mL^−1^) and PeBC1 (70.12 μg mL^−1^) solutions, respectively. Six seeds in a single pot were grown in organic soil (Flora guard SUBSTRAT). Seven-day-old chili seedlings with different concentrations of PeaT1, i.e., 80.65 μg mL^−1^ and PeBC1 and 70.12 μg mL^−1^, solutions were treated in the same way as mentioned above. The core section of the first leaves was harvested and examined, while samples were taken up to 48 h using 3.5 percent glutaraldehyde dissolved in a 0.1 M phosphate buffer (pH 7.2). All samples were cleaned in a 0.1 M phosphate buffer (pH 7.2) for about 15 min before being submerged in 1 percent osmic acid for about 2 h 5 times. For 15 min, an ethanol gradient of 100, 95, 90, 80, 70, 60, 50, and 30 percent was utilized. All critical points were dried with a Leica EM critical point dryer (CPD030; Leica Biosystems, Wetzlar, Germany). All samples were examined using a Hitachi H-7650 transmission electron microscope. To quantify the effect of PeaT1- and PeBC1-treated settlements, ten-replicates of total plant height (cm), total chlorophyll amount (SPAD), total fresh and dried weight, and the number of plant leaves were measured. The experiment was carried out three times, each time with four repetitions. The data were compared statistically using an ANOVA and LSD at α = 0.05 using a CRD randomized statistical design.

### 2.10. Plant Hormone Detection with HPLC/MS

Seeds and seven-day-old seedlings were treated as before. As previously stated, roughly 0.5 g of the seedlings’ aerial section was harvested to extract SA, JA, and ET [49]. 20 L of extraction was injected using a high-performance liquid chromatography spectrometer (HPLC/MS; Shimazu Research Instruments, ODS-C^18^, 3 m, and 2.1 per 150 mm Kyoto, Japan). HPLC was carried out at a flow rate of 0.2 mL min^−1^ with a mobile phase of 60% methanol, a column temperature of 40° C, and a sample temperature of 4° C. MS was set at the selected ion monitoring system (SIM) with a solvent temperature of 250 °C, a heat block temperature of 200 °C, a drying gas flow rate of 10 mL min^−1^, a nebulizing gas flow rate of 1.5 mL min^−1^, a detector voltage of 1.30 kV, and an interface voltage of −3.5 kV in the negative ion mode (SA m/z: 137.00; JA: 209.05). 

### 2.11. Q-RT-PCR

Kits from TransGen Biotech (Beijing, China) were used to extract RNA, synthesize cDNA, and perform a real-time quantitative polymerase chain reaction (RT-qPCR) (ABI 7500 Real-Time PCR System). An NP80 nano-photometer was used to determine the quality of RNA. The genes tested for JA, SA, and ET were (*PDF1.2*, *LOX3*, *AOS*, *ACS2*, *PAL3*, *ICS2*, *RLK*, *Thi*); the internal reference gene was ribosomal gene 18S [50], and primers for all pathways are specified (Table A1 of Appendix A). The relative fold expression of the genes was checked using the 2^−ΔΔCT^ method [51].

### 2.12. Analysis of Data

An independent Leven’s test was used to compare data from two treatments, while a two-tailed *t*-test and an LSD and an ANOVA were used to compare data from three or more treatments, respectively. For statistical data analysis, Analytical Software’s Statistix software version 8.1 (Tallahassee, FL, USA) was used. Prior to analysis, data on aphid fecundity was square-root transformed. To eliminate differences, a one-way factorial analysis of variance was performed among treatment components such as PeaT1 and PeBC1 elicitors concentrations and varied temperature regimes, followed by a 95% probability least significant difference test. The comparative CT (2^−ΔΔCT^) method was used to get gene expressions (RT-qPCR). The fold changes in the plant samples treated with the elicitor and the buffer were compared using the Student’s *t*-test (*p* = 0.05).

## 3. Results

### 3.1. M. persicae Activity 

PeaT1 and PeBC1 used two separate strategies to generate resistance to *M. persicae*, the green peach aphid. First, aphid population fall was stable in PeaT1- and PeBC1-treated chili seedlings (Table 1 and Table 2), with percentage declines in population count in PeaT1 and PeBC1 treatments compared to the buffer and control treatments. *M. persicae* preferred to feed on the control chili seedlings in the host selection experiment. The frequency of *M. persicae* colonizing PeaT1- and PeBC1-treated plants was much lower a day after the aphid was inoculated and two days after spraying the seedlings than the control, which revealed aphid colonization in regions other than buffer and PeaT1- and PeBC1-treated areas. Some aphids chose to colonize control regions over those treated with PeaT1 and PeBC1 based on their feeding habits (Figure 1A,B). Second, aphids that were fed on seedlings treated with PeaT1 and PeBC1 had a longer developing period than those that were not, whereas *M. persicae* that were fed on seedlings treated with PeaT1 and PeBC1 had a lower everyday reproductive ability (second and third nymphal instars). The second and third generations grew at a slower pace (Figure 2A,B).

### 3.2. Feeding Activity of M. persicae by EPG

The overall illustration of chili resistance variables was provided by an EPG. *M. persicae* feeding activity was considerably affected in seedlings treated with PeaT1 and PeBC1 (Table 3 and Table 4). The probing period, the length of C (pathway operation in all tissues), and the sum of *M. persicae* Pd (potential decrease in cell punctures) in the PeaT1- and PeBC1-treated chilli seedlings were significantly reduced, whereas the period of non-probe time before the first E (phloem-feeding activity) and the total duration of F (penetration problems) increased significantly. During the non-probing period, there was no electrical contact between the aphid stylet and the plant. The non-probing period before the first E was noticeably improved in the PeaT1 and PeBC1 treatments, implying a repellent or deterrent surface feature in the PeaT1 and PeBC1-treated chilli seedlings. C waves show intercellular type motion and may act as a mechanical plant barrier. The shorter the C waves (<3 min) detected, the greater the mechanical difficulty in seedlings treated with PeaT1 and PeBC1. Additionally, a decreased Pd number (cell puncture) was linked to aphid resistance in plants, which could be attributed to mechanical difficulties (the PeaT1- and PeBC1-treated chili seedlings in present study). Wave E1 indicated aphid saliva injection during phloem-feeding activities into sieve elements. In contrast, the E2 wave (sap sucking during phloem-feeding activities) showed phloem sap injection with concurrent salivation, which could have reflected a mesophyll or vascular resistance factor. In the sieve element, an extended E1 indicated more plugging or defense compounds. There was, however, no substantial difference between the control and PeaT1 and PeBC1 treatments in the E2 period, indicating no or low variability in phloem compounds to confer resistance to *M. persicae*. However, the period of the F wave in the PeaT1- and PeBC1-treated chili seedlings was higher, indicating that PeaT1 and PeBC1 induced an enhanced mechanical defense. The EPG results suggested that the resistance induced by PeaT1 and PeBC1 was mainly due to the modification of physical defenses.

### 3.3. Influence of PeaT1 and PeBC1 Elicitor on Aphid’s Nymphal Development Time

Factorial analysis showed an impact on the overall developmental time of *M. persicae* on various PeaT1 and PeBC1 concentrations in three different temperature regimes and their interactions. A differential trend was identified in the developmental time of nymphs for the elicitor effect at a different temperature, as shown in Table A2 and Table A3 of Appendix B. The development time of each nymphal instar was extended when the concentrations of PeaT1 (Figure 3) and PeBC1 (Figure 4) increased. At a low temperature of 23 °C, the maximum development time for the fourth nymphal instar was 3.9 days at a high concentration (80.65 µg mL^−1^ and 70.12 µg mL^−1^, respectively). A minimum nymph growth time was attained for the first instar at a low concertation (12.09 µg mL^−1^ and 10.51 µg mL^−1^, respectively) at a high temperature of 27 °C. The buffer control and water-treated control have different times for nymph development. In general, nymphal development took longer at low temperatures than at medium or high temperatures for all instars. A maximal extension of time was recorded at the fourth instar for each concentration of the elicitor with varying temperature regimes. PeaT1 and PeBC1, demonstrated a superior importance for the first, second, third, and fourth instar aphid concentrations of the elicitors. The effect of temperature regimes on the developmental period of nymphs in the first, second, third, and fourth instar aphids was also significant. The nymphal aphid development time showed little fluctuation across their shared interface after that, and the nymphal development period revealed no modifications in terms of joint interfaces.

### 3.4. Effect of PeaT1 and PeBC1 Elicitor on the M. persicae Fecundity

Figure 5 and Figure 6 show that the aphid fecundity was significantly affected by different concentrations of PeaT1 and PeBC1 and temperature regimes, as shown in Table A4 and Table A5 of Appendix C. Figure 5 and Figure 6 show that *M. persicae* produced fewer offspring than those that were fed on (water) positive and (buffer) negatively treated chili plants in contrast to the PeaT1- and PeBC1-treated seedlings. However, a minimum fecundity of aphid at a maximum temperature was observed at 27 °C in seedlings treated with elicitor proteins PeaT1 and PeBC1, and a maximum fecundity of aphid at a minimum temperature of 23 °C was recorded in seedlings treated with elicitor proteins PeaT1 and PeBC1.

### 3.5. Effect of PeaT1 and PeBC1 on the Growth and Structure of Chili

PeaT1 and PeBC1 significantly altered plant height and surface assembly of chili leaves (Figure 7A,B). SPAD, total number of leaves, and fresh and dry weight all showed a similar pattern. The height of the plant was higher in PeaT1- and PeBC1-treated seedlings than in control seedlings. PeaT1 and PeBC1 elicitor proteins dramatically altered the surface structures in terms of trichomes formation on chilli leaves, and seedlings treated with PeaT1 (56.23 ± 0.42 mm^−2^) and PeBC1 (52.14 ± 0.34 mm^−2^) had more trichomes than control (30.17 ± 0.16 mm^−2^). A more complex wax structure was created, resulting in a much-improved surface environment. This is a characteristic that is thought to be detrimental to aphid colonization.

### 3.6. SA, JA, and ET Accumulation in PeaT1- and PeBC1-Treated Chili Seedlings

To analyze the relationships between SA, JA and ET with cuticular wax deposition and increases in the trichomes density of PeaT1, the aphid infestations or both were analyzed. PeaT1 and PeBC1 showed a high accumulation of JA, SA, and ET in seedlings (Figure 8A,B); all three signaling pathways were found to participate in aphid-induced chili resistance. Furthermore, in PeaT1- and PeBC1-treated plants, JA, SA, and ET accumulated, implying that protein elicitors at least partially activated the defense response in chili plants. JA, SA, and ET induction are acknowledged to be affected by aphid numbers, infestation rates, and aphid species.

### 3.7. Relative Fold Changes in the Expressions of Defense-Related Genes

PeaT1 and PeBC1 enhanced the defense mechanism in chili seedlings. *PDF1.2* (plant defensin 1.2), *LOX3* (lipoxygenase multifunctional proteins), *AOS* (Allene oxide synthase), *ACS2* (ACC synthase 2), *PAL3* (Phenylalanine ammonia-lyase), *ICS2* (Isochorismate synthase 2), *RLK* (Receptor-like kinase), and *Thi* (Thionin) were selected as test genes for defense pathways. Treatment with PeaT1, aphid infestation (Figure 9), and PeBC1 (Figure 10) elevated all genes; the transcripts of all genes were statistically larger with PeaT1 and PeBC1 than with the other two treatments. Treatment with PeaT1 and PeBC1 was thought to boost induced resistance to aphid infestation. The improved transcript of JA and SA test genes also revealed their functions in the JA and SA pathways in wheat aphid resistance, as Moron and Thompson demonstrated. LOX3 was the most highly expressed gene, followed by ACS2, ICS2, PAL3, and PDF1.2, Thi, RLK, and AOS, in that order. The transcription of the test genes was found to be the cause of aphid resistance.

## 4. Discussion

The use of elicitors is a new biological control tool for the management of insect pests, as they play a dynamic role in the defense and signaling mechanisms of plants under attack by sap-feeding insects [27,28,29,52]. In order to assess its potential role against green peach aphids *M. persicae*, this work undertook an in vitro evaluation of elicitor proteins PeaT1 and PeBC1, isolated from fungal strains of *A. tenuissima* and *B. cinerea*, respectively. Several elicitor protein strains have been shown to have a wide range of antimicrobial properties, acting as antimicrobial peptides in bacteria and fungi. By mixing DNA and RNA, they can enter the cell and transfer to the cytoplasm and nucleus, disrupting protein synthesis [53]. Pathogenic bacteria and fungi, whether necrotrophic or biotrophic, are a substantial source of elicitors such as PAMPs and MAMPs [54]. In this investigation, the potential activity of PeaT1 and PeBC1 for *M. persicae* treatment was demonstrated. Chemical elicitors, such as methyl-jasmonate, benzothiadiazole, and other plant defenses, such as proteinase inhibitors, have been established to significantly suppress the activity of herbivorous pests in tomato crops in earlier research [54]. The application of a methyl salicylate elicitor reduced the soybean aphid *A. glycines* by up to 40%, according to the discoveries of present research study [55]. Bioassays revealed that population development was substantially slower on PeaT1- and PeBC1-treated chilli plants compared to buffer and control plants. Exogenous applications of elicitors, such as MJ, JA, and BTH, have been demonstrated to have a deleterious impact on the population growth and fitness of various aphid species in previous investigations, which is supported by the current findings [55]. Biocontrol potential has also been revealed for a variety of Diptera, Coleoptera, and Lepidoptera, as well as nematodes and mollusks [56]. PeaT1 and PeBC1 were found to have the ability to inhibit herbivores by influencing population and growth characteristics in the current investigation. Physical resistance to harmful bacteria and herbivores is provided by trichomes. In Solanum spp., these hairy adjuncts of plant epidermal cells impact herbivore morphology and the density role of trichomes, i.e., seven trichomes with two key defense-related effects were investigated [57]. For starters, a plant’s surface acts as a physical barrier since its dense matte hair supplies energy, restricts feeding capacity, and prevents insects from reaching the surface [58]. *M. persicae* avoids plants with a lot of hair, such Solanum hirsutum. Trichomes are also linked to the tomato plant’s basic defensive mechanism, as the surface area covered by epidermal cell appendages of unicellular or multicellular hairs provides resistance to a range of pests as a result of the plant’s “pubescence.” Leaf beetle (Coleoptera: Chrysomelidae) settlement was reduced in soybeans with thick trichomes compared to plants with trichomes removed, which attracted more beetles [59,60]. The EPG results revealed that PeaT1 and PeBC1 resistance was mostly related to a change in physical defences [36,42,47,61,62,63]. By activating a photosynthetic pathway that paralleled plant development characteristics, PeaT1 and PeBC1 reduced disease severity and improved induced resistance in PeaT1- and PeBC1-treated chilli seedlings [64,65].

The PeaT1- and PeBC1-treated seedlings and leaves had more trichomes than the controls. Aphids with an increased number of trichomes were reported to be hindered from reproducing and settling on PeaT1 and PeBC1-treated chilli seedlings and leaflets. A high density of trichomes had a negative impact on the feeding activity of Leptinotarsa decemlineata. The cell wall, which underlies plant resilience and is a sign of advances in the cell wall, is another important component of the physical barrier lignin [64,66,67]. An increase in lignin concentration in Chrysanthemum improved aphid tolerance [68]. Plants use trichomes and wax production as physical defenses in response to biotic and abiotic stress. Direct damage, such as leaf-cuts, methoxyfenozide, and manganese, can lead to their establishment [69]. Exogenous phytohormones, MJ, or JA, as demonstrated in *Arabidopsis* and tomatoes, can also influence cuticular wax deposition and trichome density [57,70]. Brassica napus wax deposition necessitated the use of SA [71]. SA and JA accumulations in PeaT1- and PeBC1-treated chili plants can thus be linked to enriched trichome compactness and cuticular wax confession, respectively [13,72,73]. In addition, the treatment of the PeaT1 and PeBC1 elicitors had adverse effects on aphid fecundity. Aphids produce fewer offspring on PeaT1- and PeBC1-treated plants/seedlings as compared to water, and buffer treated ones. The results were consistent with previous studies showing that exogenous SA and MJ can cause lower mean lifetime fecundity in aphids [54]. Therefore, optimum temperatures (e.g., 25 °C) demonstrated a maximum aphid fecundity, with the minimum fecundity at higher temperatures (27 °C) due to a decreased metabolic rate [74]. Similarly, a variance analysis showed that, in PeaT1- and PeBC1-treated plants, the development time of nymphs was extended compared to the control; even at a lower temperature (23 °C), the maximum nymphal development time was observed, indicating that a one-degree temperature increase affected the life cycle of the insects [75]. Additional studies need to be conducted to understand the underlying mechanism of PeaT1 and PeBC1 on chili plants, particularly the inhibitory effect of both elicitors on nymphal development time and fecundity. 

Additionally, JA, SA, and ET increased marker gene transcriptions, signaling that they play an essential role in chili’s resistance to aphids. After aphid infestation in *Arabidopsis*, the transcript genes *PDF1.2* (plant defensin 1.2), *LOX3* (lipoxygenase multifunctional proteins), *AOS* (Allene oxide synthase), *ACS2* (ACC synthase 2), *PAL3* (Phenylalanine ammonia-lyase), *ICS2* (Isochorismate synthase 2), *RLK* (Receptor-like kinase 2), and *Thi* (Thionin) were significantly increased. Actin is a structural component in the plant cell wall that is depolymerized via the regulation of cell and cross-linking [76,77]. Actin depolymerization is negatively related to aphid fecundity and population [78]. JA, SA, and ET genes impart resistance to insect herbivorous diseases and pathogens, which enhances plant defense responses [10,79,80]. All JA, SA, and ET test genes showed significant and robust regulation [15]. *PAL3* coding for ammonia-lyase phenylalanine is involved in cell wall construction, as demonstrated in *Arabidopsis* [81]. *PDF 1.2* coding for plant defense in 1.2 is a predominant JA-signaling effector and is significantly induced in many plants following infection with necrotrophic pathogens [82]. *LOX3* coding for lipoxygenase multifunctional proteins lipoxygenase, JA, and *LOX1*, makes upregulation occur in chili plant *Pseudomonas* following injection [14,82,83,84,85,86]. Increased transcription of *AOS* (allene oxide synthase) improved aphid resistance, as shown in tomato by Thompson and Moran [79]. *ACS2* is the rate-limiting enzyme that regulates ET biosynthesis for pathogen attack responses. Positive control of *ACS* genes by the *MAPK* signaling cascades contributes to an increased production of ET in plants when challenging necrotrophic fungal pathogens [87]. *ICS2* coding for isochorismate synthase during the production of SAR is involved in sustained SA synthesis [88]. Many lines of evidence have suggested different routes for the mechanism of SAR signaling in plants. Studies in search of SAR’s systemic signal show that SA acts as a systemic signal in SAR [88]. *RLK* coding for receptor-like kinases (RLKs) involves a variety of plant responses, including development, growth, hormone perception, and response to pathogens. *RLKs* and *RLPs* are at the heart of the plant’s early warning system following pathogen attacks. *RLKs* have been identified as having a function in broad-spectrum, elicitor-initiated protection responses, and as dominant resistance (*R*) genes in race-specific pathogen defense mechanisms [89,90,91,92]. Plants deploy a large number of *RLKs* and *RLPs* as pattern recognition receptors (*PRRs*) that detect molecular patterns derived from microbes and host the first layer of inducible protection [93]. *Thi* coding for Thionin, a low, highly specific cysteine-rich protein with antimicrobial activity, is known to contribute to plant defense against many bacteria and fungi [94,95]. The role of γ-thionin in providing defense against a wide range of fungi has been elucidated [96], and an elevated level of expression of the *PDF1.2* gene (which encodes for defensin) was observed in TMV-infected *N. benthamiana* plants [97]. The enhanced expression of γ-thionin therefore suggests a possible role of this protein in imparting basal defense responses in resistant chili plants. Findings from this study confirm the activation by *M. persicae* of JA, SA, and ET pathway-associated genes [98].

## 5. Conclusions

In this paper, we present data on aphid resistance in chili with the prolonged developmental time of the first to the fourth nymphal instars that are related to a lower fecundity of *M. persicae*. Increased PeaT1 and PeBC1 concentrations were found to affect aphid colonization. Resistance factors were verified by the increased number of trichomes and wax amounts, which were mainly involved in mechanical defenses. Likewise, an EPG study presented and gave an idea that the resistance induced by PeaT1 and PeBC1 might be due to the modification of physical defense and an increased number of trichomes, while wax composition and antibiotic or antixenosis effects affected aphid feeding behavior in PeaT1- and PeBC1-treated seedlings. Moreover, our study focused on the effect of PeaT1 and PeBC1 on the growth and structure of chili, and we found that increased plant height and modified surface structures of the chili leaves were greatly influenced by PeaT1 and PeBC1. We also confirmed the role of PeaT1 and PeBC1 in physical defense against *M. persicae*. The physical defense response induced by PeaT1, PeBC1, JA, SA, and ET contributed to a comprehensive plant physical response. However, some issues need to be resolved in the future, e.g., “how JA, SA, and ET work to induce resistance,” and “whether or not other plant hormones are involved.” Nevertheless, the current study showed that PeaT1 isolated from *A. tenuissima* and PeBC1 from *B. cinerea* could be applied to chili seeds and seedlings to protect plants against *M. persicae* Sulzer.

## Figures and Tables

**Figure 1 microorganisms-09-02197-f001:**
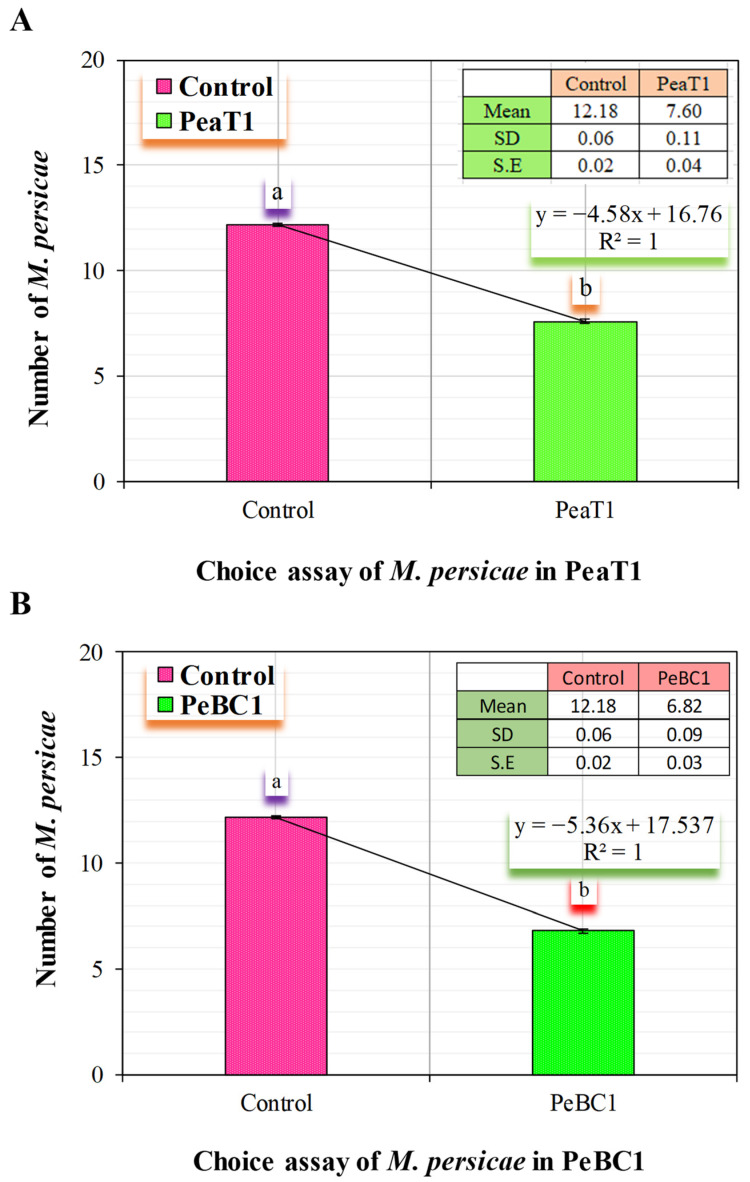
*M. persicae* feeding preferences (**A**) *M. persicae* preferred feeding on control plants in PeaT1; (**B**) *M. persicae* feeding on PeBC1-treated and control-treated chilli seedlings 24 h after infestation colonization (mean ± SD). SPSS 18.0 was used to compare data using one-way analysis of variance (ANOVA) and the least significant difference (LSD). Lower style alphabet letters show significant variations across all treatments. (*p* = 0.05).

**Figure 2 microorganisms-09-02197-f002:**
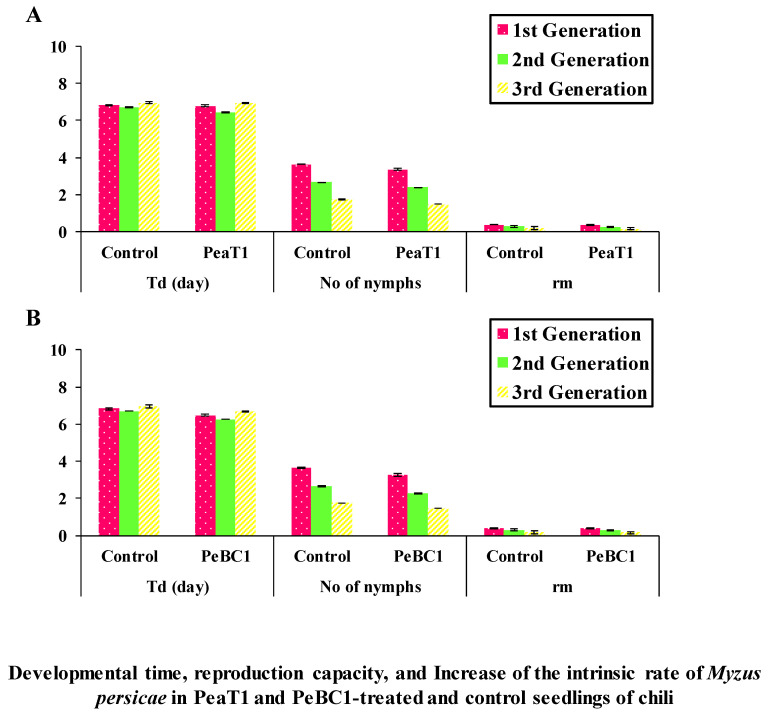
In PeaT1- and PeBC1-treated (**A**,**B**) and control seedlings of chili, developmental time, reproductive capacity, and growth of the intrinsic rate of *M. persicae* are reported as mean ± SD. *Td* stands for development period, nymphs per day stands for average reproduction ability, and *r_m_* stands for intrinsic rate rise. To compare data, SPSS 18.0 was utilized with one-way analysis of variance (ANOVA) and the least significant difference (LSD) (*p* = 0.05).

**Figure 3 microorganisms-09-02197-f003:**
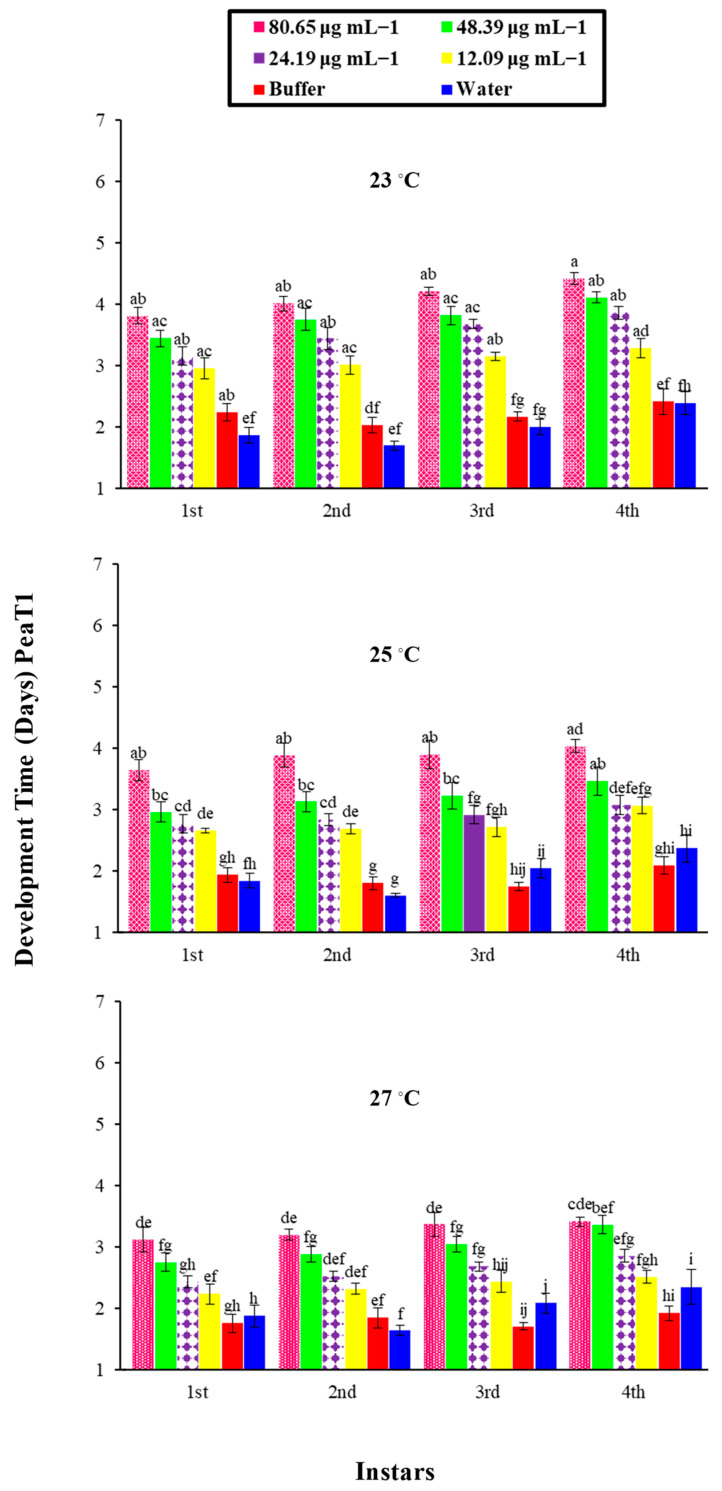
Mean developmental time, (±SE) of different nymphal instars of *M. persicae* on chili plants by PeaT1 elicitor protein at different concentrations and different temperature (23, 25, 27 °C) regimes (*n* = 10). Different alphabets above bar tops specify significant differences between treatments (factorial analysis one way ANOVA; LSD at α = 0.05).

**Figure 4 microorganisms-09-02197-f004:**
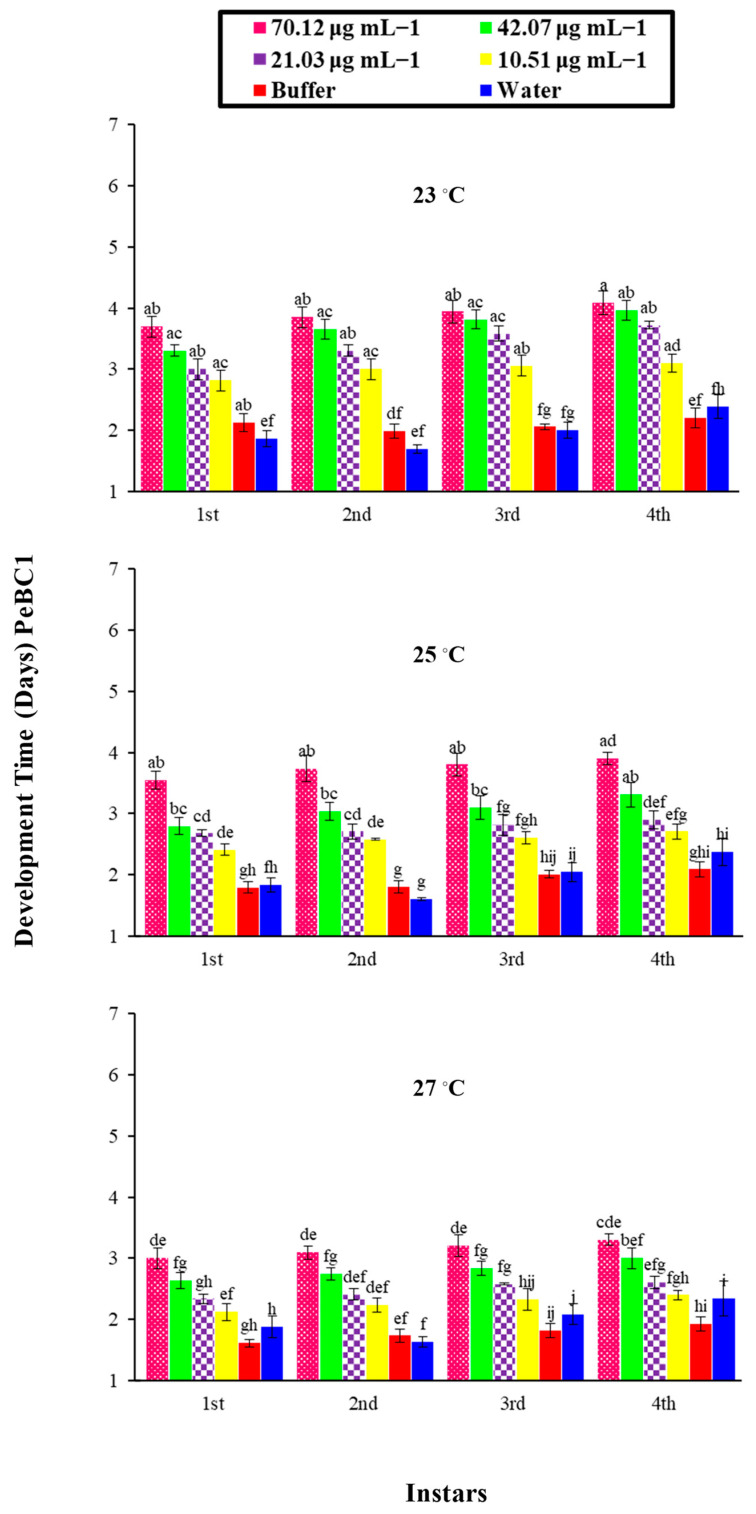
Mean developmental time (±SE) of different nymphal instars of *M. persicae* on chili plants by PeBC1 elicitor protein at different concentrations and different temperature (23, 25, 27 °C) regimes (*n* = 10). Different alphabets above bar tops specify significant differences between treatments (factorial analysis one way ANOVA; LSD at α = 0.05).

**Figure 5 microorganisms-09-02197-f005:**
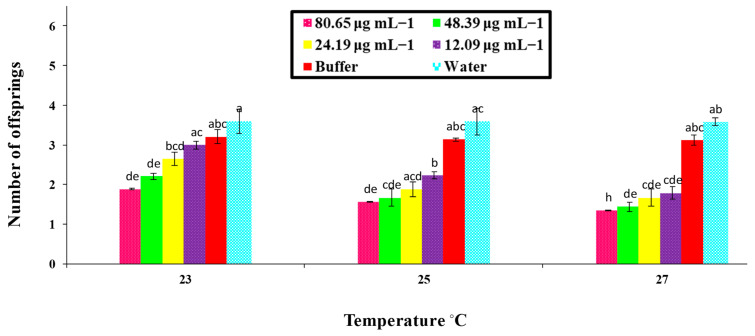
Average fecundity with (±SE) of aphids (*M. persicae*) on chili plants with various PeaT1 elicitor protein changed concentrations at different altered temperature regimes (*n* = 10); alphabet letters on each bar’s top show inequalities between treatments (factorial analysis one-way ANOVA; LSD at α = 0.05). Fecundity was reduced in PeaT1-treated seedlings.

**Figure 6 microorganisms-09-02197-f006:**
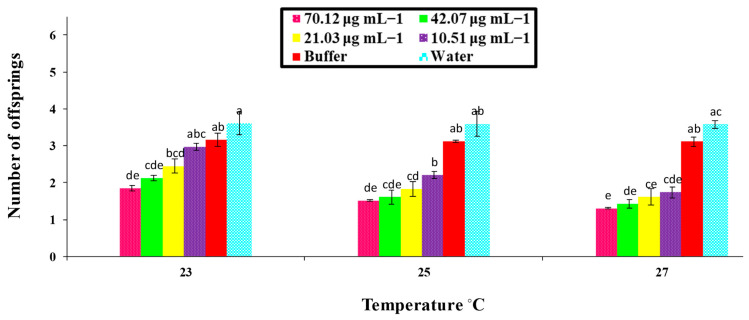
Average fecundity with (±SE) of aphids (*M. persicae*) on chili plants with various PeBC1 elicitor protein changed concentrations at different altered temperature regimes (*n* = 10); alphabet letters on each bar’s top show inequalities between treatments (factorial analysis one-way ANOVA; LSD at α = 0.05). Fecundity was reduced in PeBC1-treated seedlings.

**Figure 7 microorganisms-09-02197-f007:**
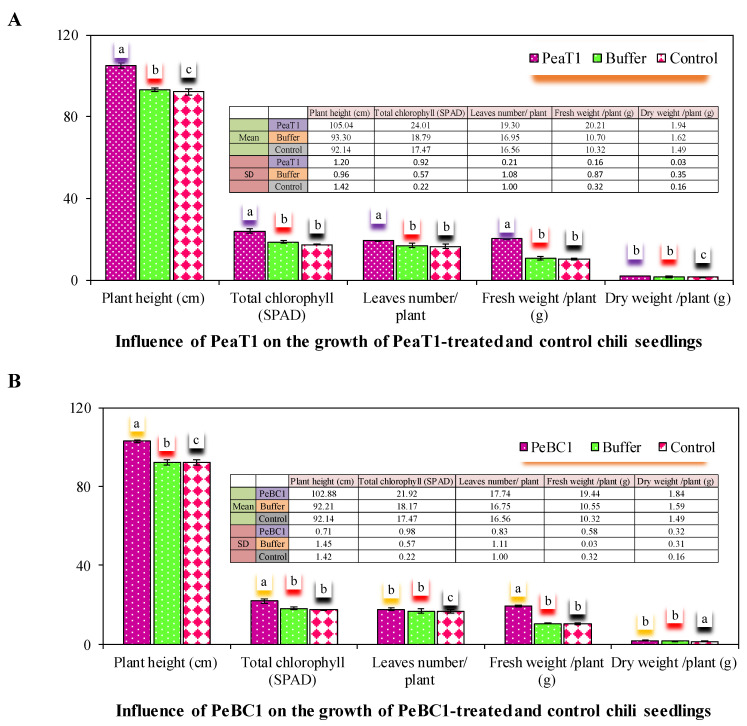
Influence of PeaT1 and PeBC1 on the growth of treated chili seedlings (**A**,**B**). Average growth parameters with (±SE) of chili plants in PeaT1, PeBC1, buffer, and control-treated seedlings (*n* = 10). Statistically, data were compared by the least significant difference (LSD), one-way analysis of variance (ANOVA), and Levene’s test in SPSS 18.0. Different lower style alphabets letters indicate a significant difference in different treatment patterns (*p* = 0.05).

**Figure 8 microorganisms-09-02197-f008:**
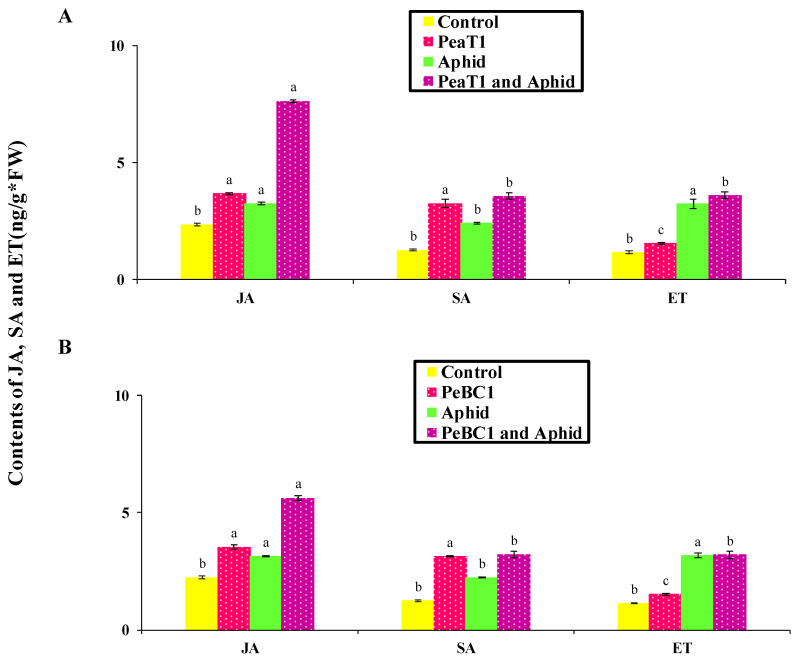
Chili seedlings’ jasmonic acid (JA), salicylic acid (SA), and ethylene (ET) levels (mean ± SD). (**A**,**B**), respectively. One day after spraying, data on PeaT1 and PeBC1 treatment was collected. The aphids were inoculated one day after seedlings were sprayed in both treatments, and samples were collected one day later. The LSD, ANOVA, and Leven’s test were used to compare data statistically. Version 8.1 of Statistix. Lower-case letters indicate significant differences between various treatments in JA, SA, or ET (*p* = 0.05).

**Figure 9 microorganisms-09-02197-f009:**
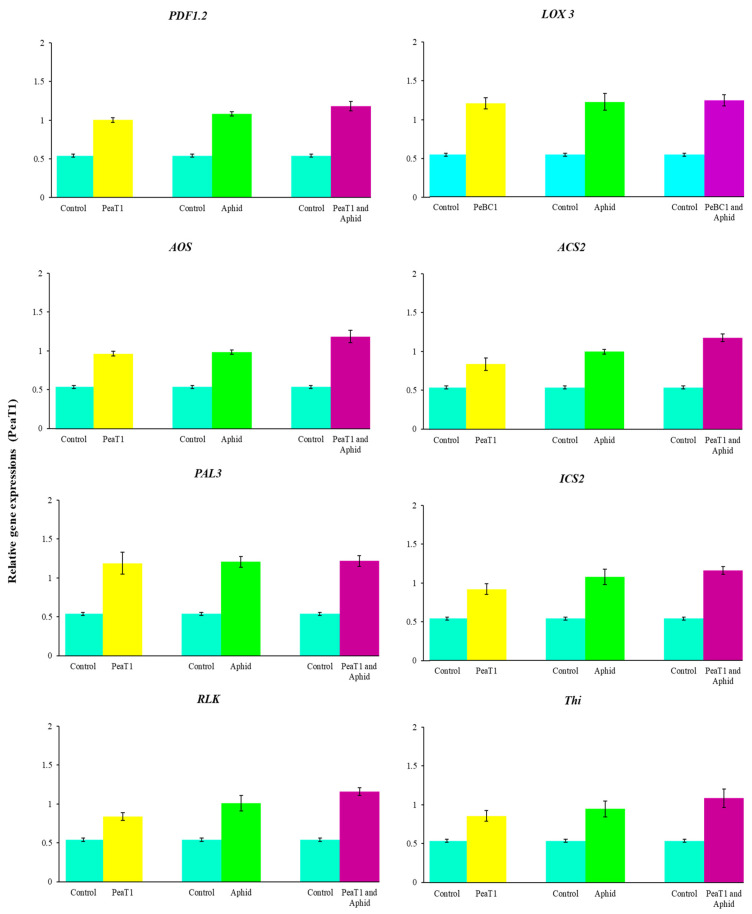
Treatment with PeaT1-elicitor, PeaT1 with aphid, and aphid infestation alone revealed relative expression of the JA, SA, and ET pathway genes. One day after spraying, PeaT1 treatment was carried out. The aphids were inoculated one day after the seedlings were sprayed in both treatments, and the samples were collected one day later. Using SPSS 18.0, data were compared using LSD, one-way ANOVA, and Levene’s test. The fold expression of treated chilli seedlings is represented by bars. Controls were given a relative fold expression of 0.54.

**Figure 10 microorganisms-09-02197-f010:**
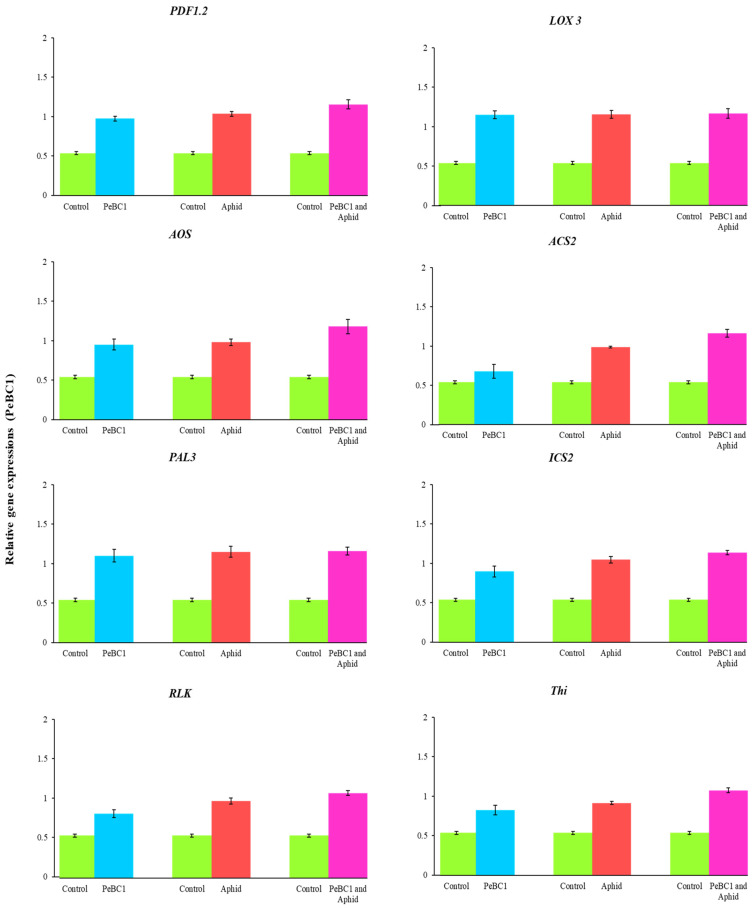
Treatment with PeBC1-elicitor, PeBC1 with aphid, and aphid infestation alone revealed relative expression of the JA, SA, and ET pathway genes. One day after spraying, PeaT1 treatment was carried out. The aphids were inoculated one day after the seedlings were sprayed in both treatments, and the samples were collected one day later. Using SPSS 18.0, data were compared using LSD, one-way ANOVA, and Levene’s test. The fold expression of treated chilli seedlings is represented by bars. Controls were given a relative fold expression of 0.54.

**Table 1 microorganisms-09-02197-t001:** *M. persicae* population variations were seen in PeaT1-, control-, and buffer-treated chili seedlings. To compare data, one-way analysis of variance (ANOVA), Levene’s test with SPSS 18.0, and the least significant difference (LSD) were employed. After aphid inoculation on the same day, significant variations in the letters in the rows can be noticed in all treated samples (*p* = 0.05).

Days after Aphid Inoculation	Control	Buffer	PeaT1
5	53.91 ± 0.05 ^b^	58.24 ± 0.01 ^a^	45.23 ± 0.02 ^c^
10	108.21 ± 0.04 ^b^	124.26 ± 0.04 ^a^	86.24 ± 0.05 ^c^
15	214.31 ± 0.06 ^b^	249.17 ± 0.05 ^a^	178.26 ± 0.02 ^c^

**Table 2 microorganisms-09-02197-t002:** *M. persicae* population variations were seen in PeBC1-, control-, and buffer-treated chili seedlings. To compare data, one-way analysis of variance (ANOVA), Levene’s test with SPSS 18.0, and the least significant difference (LSD) were employed. After inoculation of aphid on the same day, significant changes in letters in rows can be noticed in all treated samples (*p* = 0.05).

Days after Aphid Inoculation	Control	Buffer	PeBC1
5	53.91 ± 0.05 ^b^	57.40 ± 0.23 ^a^	43.12 ± 0.03 ^c^
10	108.21 ± 0.04 ^b^	123.35 ± 0.03 ^a^	84.57 ± 0.04 ^c^
15	214.31 ± 0.06 ^b^	248.15 ± 0.03 ^a^	175.34 ± 0.06 ^c^

**Table 3 microorganisms-09-02197-t003:** *M. persicae* electrical penetration graph (EPG) data on PeaT1-treated and untreated chilli plants. Mean ± SD. Pathway activities are represented by C, potential drop is represented by Pd, phloem-feeding E represents activities, F represents penetration difficulty, G represents xylem-feeding activities, saliva injection is represented by E1, and sap sucking is represented by E2. Data were compared statistically using an independent *t*-test with two tails in SPSS 18.0. The difference between PeaT1 and control treatment with the same parameters of * (*p* = 0.05) is shown by asterisks.

EPG Parameters	Control (*n =* 20)	PeaT1 (*n =* 20)
Total probing time (h)	3.78 ± 0.05	2.96 ± 0.01
Number of C	16.45 ± 0.04	26.73 ± 0.04 *
Number of short probes (C < 3 min)	9.12 ± 0.07	24.12 ± 0.16
Duration of non-probe period before the 1st E (h)	3.92 ± 0.06	3.89 ± 0.06 *
Number of pd	72.87 ± 0.05	36.42 ± 0.07
Mean duration of Pd(s)	8.14 ± 0.04	7.69 ± 0.09
Number of E1	5.23 ± 0.05	4.42 ± 0.07
Mean duration of E1(min)	8.91 ± 0.04	10.12 ± 0.07
Number of E2	0.88 ± 0.07	0.73 ± 0.07 *
Mean duration of E2 (h)	29.96 ± 0.10	43.78 ± 0.05
Number of G	0.86 ± 0.06	0.79 ± 0.08
Mean Duration of G (min)	19.14 ± 0.04	14.67 ± 0.05
Number of F	5.24 ± 0.05	3.13 ± 0.06
mean duration of F (min)	22.24 ± 0.03	52.97 ± 0.04

**Table 4 microorganisms-09-02197-t004:** *M. persicae* electrical penetration graph (EPG) data on PeBC1-treated and untreated chilli plants. Mean ± SD. Pathway activities are represented by C, potential drop is represented by Pd, phloem-feeding E represents activities, F represents penetration difficulty, G represents xylem-feeding activities, saliva injection is represented by E1, and sap sucking is represented by E2. Data were compared statistically using an independent *t*-test with two tails in SPSS 18.0. The difference between PeBC1 and control treatment with the same parameters of * (*p* = 0.05) is shown by asterisks.

EPG Parameters	Control (*n =* 20)	PeBC1 (*n =* 20)
Total probing time (h)	3.14 ± 0.02	2.12 ± 0.01 *
Number of C	15.78 ± 0.77	25.66 ± 1.61 *
Number of short probes (C < 3 min)	8.67 ± 0.81	23.43 ± 1.21 *
Duration of non-probe period before the 1st E (h)	3.76 ± 0.13	3.87 ± 0.03
Number of pd	72.18 ± 0.05	35.15 ± 0.03
Mean duration of Pd(s)	7.71 ± 0.06	7.23 ± 0.15
Number of E1	4.14 ± 0.02	3.45 ± 0.08
Mean duration of E1(min)	8.79 ± 0.07	9.67 ± 0.08 *
Number of E2	0.64 ± 0.05	0.54 ± 0.02
Mean duration of E2 (h)	29.74 ± 0.02	43.27 ± 0.04
Number of G	0.74 ± 0.06	0.68 ± 0.13
Mean Duration of G (min)	18.23 ± 0.05	13.57 ± 0.01
Number of F	4.67 ± 0.06	2.43 ± 0.03
mean duration of F (min)	21.46 ± 0.02	± 0.03

## Data Availability

The required data set is already available in manuscript file, other data sets generated during the study are available upon request from corresponding author.

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
