# Peer review of "PeaT1 and PeBC1 Microbial Protein Elicitors Enhanced Resistance against Myzus persicae Sulzer in Chili Capsicum annum L."

_microorganisms, 2021, doi:10.3390/microorganisms9112197_

Round 1
Reviewer 1 Report
After the review, the MS “Potential use of Electrical Penetration Graph (EPG) to induce resistance in Chilli (Capsicum annum) plant against Green peach Aphid (Myzus persicae) by Microbial Protein Elicitors PeBC1 obtained from Botrytis cinerea and PeaT1 from Alternaria tenuissima” has been improved, however, it still needs further corrections.
Major remarks
Too much attention has been paid to the importance of trichomes in plant resistance to aphid feeding in Discussion, while the issue is extremely sparingly presented in the Results section.
Since the Results and the Discussion are separate chapters, do not include references to the literature in the results section (e.g. line 565, 568-69, 591, etc.).
Full Latin names should be used only for the first mention, the abbreviation should be used in the following text (e.g. Myzus persicae Sulzer for the first mention and M. persicae throughout the manuscript Line: 70, 83, 85, 95, 113, 208… . The same applies to the full names of chemical/biochemical compounds.
Detailed remarks
Keywords: The study investigated plant resistance not aphid resistance (more precisely: induced plant resistance against to aphids).
L 145: iron chlorosis is duplicated with different references
L 162 and later: green peach aphid (this is the common name of the species) not green peach aphids
L 27, 31, 33, 39, 45 (throughout MS) : use only common or Latin name (M. persicae) without “Sulzer”
L 181, 337: use % instead percent;
L 190-195: The sentence “The Äkta Explorer Protein Purification System (Amersham Biosciences, Temecula, CA, USA), as described by Wang et al. [31] with a His-Trap HP column (GE Healthcare, Waukesha, WI, USA), used various loading buffers (A, B, C, and D) for the further purification of elicitor protein PeBC1.” is not clear.
L 185-228: Combine the same steps according to “Expression and purification of PeaT1 and PeBC1” and only express the differences.
L 275: Petri dish
L 324, 348: use “α”
L 389: remove “indoors”
L 445: stylet not style
L 446-448: the sentence is hard to follow.
L 487: “… time was 3.9 d at a low…” what is d?
L 492: longer not higher
L 493: “… temperature of the elicitor….” ?
L 671: start “Chrysanthemum” with capital letter
L 494-496: The sentence is hard to follow.
Figure 3 and 4: M. persicae without parenthesis. For what purpose the marks a, b, c were placed in the caption if they are not exist on the chart? What is BSA meaning?
Figure 5: Do not put “PeaT1” in the axis name; remove “with (a, b) ± SE” from the caption; check the correctness of the letters above bars at 27°C
L 522-524: The sentence is confusing. Did the authors wish to express that the aphid fecundity was significantly affected by different concentration of PeaT1 and PeBC1 and temperature regimes?
L 527-530 “However, a minimum fecundity at a maximum temperature was observed at 27° C, and a maximum fecundity at a minimum temperature of 23 °C was recorded.” In my opinion it is true only for seedlings treated by elicitors (Figure 5 and 6).
L 37-39; 548-550: There are any data about trichomes and wax structure on seedlings (any photo. table, figure?). The finding “more trichomes” is not a research result.
Figure 7: What is PeBL1 in the caption?
Figure 8: Check the letters above bars please, are they put correctly?
Figure 9 and 10: word sequences are repeated
L 621: tenuissima
L 660-663: The sentence is hard to follow.
L 683: Plants couldn’t produce aphids! Aphids produce fewer offspring on PeaT1 and PeBC1 treated plants/seedlings as compared to water and buffer treated ones.
L 690: “on” plants not “in” plants
L 687-694: The effect of temperature on aphid development is well known. The authors probably wanted to demonstrate the inhibitory effect of the studied elicitors on the development and fertility of aphids regardless of temperature. Thus the sentence should be reconstructed.
L 755: EPG study doesn’t confirm resistance due to strict (e.g. trichomes) physical features or chemical composition of plant. It only show the aphid feeding mode. It is not possible to say unequivocally what factor caused the feeding difficulties.

Author Response
Manuscript ID microorganisms-1421498
Reviewer 1 Comments
Question 1: Since the Results and the Discussion are separate chapters, do not include references to the literature in the results section (e.g. line 565, 568-69, 591, etc.)?
Answer: Dear reviewer the authors are thankful for your patience and guidance towards this manuscript, all the references from the results section have been removed as per your valuable suggestions
Question 2: Full Latin names should be used only for the first mention, the abbreviation should be used in the following text (e.g. Myzus persicae Sulzer for the first mention and M. persicae throughout the manuscript Line: 70, 83, 85, 95, 113, 208, The same applies to the full names of chemical/biochemical compounds?
Answer: Dear reviewer the authors are thankful for your valuable suggestions towards this manuscript, the recommended suggestions have been followed
Question 3: Keywords: The study investigated plant resistance not aphid resistance (more precisely: induced plant resistance against aphids?
Answer: The recommended suggestion has been followed and highlighted in the revised version of the manuscript
Question 4: L 145: iron chlorosis is duplicated with different references
Answer: The duplicated text has been removed and highlighted in the revised version of the manuscript i.e. iron chlorosis [40],
Question 5: L 27, 31, 33, 39, 45 (throughout MS): use only common or Latin name (M. persicae) without “Sulzer”
Answer: The recommended suggestions have been followed and highlighted in the revised version of the manuscript
Question 6: L 181, 337: use % instead percent;
Answer: Percent has been replaced with % and highlighted in the revised version of the manuscript
Question 7: L 190-195: The sentence “The Äkta Explorer Protein Purification System (Amersham Biosciences, Temecula, CA, USA), as described by Wang et al. [31] with a His-Trap HP column (GE Healthcare, Waukesha, WI, USA), used various loading buffers (A, B, C, and D) for the further purification of elicitor protein PeBC1.” is not clear?
Answer: The sentence structure has been restructured in the revised version of the manuscript i.e.
The elicitor proteins PeaT1 and PeBC1 were purified using the Akta Explorer Protein Purification System (Amersham Biosciences, Temecula, CA, USA) and a His-Trap HP column (GE Healthcare, Waukesha, WI, USA) with various loading buffers (A, B, C, and D) as described by Wang et al. [31].
Question 8: L 185-228: Combine the same steps according to “Expression and purification of PeaT1 and PeBC1” and only express the differences.
Answer: The expression and purification steps have been combined for both PeaT1 and PeBC1 in the revised version of the manuscript
Question 9: L 275: Petri dish, L 324, 348: use “α”
Answer: The recommended suggestions have been followed and highlighted in the revised version of the manuscript
Question 10: L 389: remove “indoors”
Answer: The recommended suggestion has been followed and highlighted in the revised version of the manuscript
Question 11: L 446-448: the sentence is hard to follow.
Answer: These lines have been restructured more clearly in a revised version of the manuscript i.e.
Additionally, a decreased Pd number (cell puncture) was linked to aphid resistance in plants, which could be attributed to mechanical difficulties (the PeaT1- and PeBC1-treated chilli seedlings in the present study). Wave E1 indicated aphid saliva injection during phloem-feeding activities into sieve elements.
Question 12: L 487: “… time was 3.9 d at a low…” what is d?
Answer: d represents a day
Question 13: L 492: longer, not higher
Answer: The word has been revised as per your suggestions in the revised version of the manuscript
Question 14: L 493: “ temperature of the elicitor….”?
Answer: The line has been modified in the revised version of the manuscript i.e.
at each concentration of the elicitor with different temperature regimes,
Question 15: L 671: start “Chrysanthemum” with a capital letter
Answer: c has been replaced with C in a revised version of the manuscript
Question 16: L 494-496: The sentence is hard to follow.
Answer: The sentence has been modified more clearly in the revised version of the manuscript i.e.
Following that, the nymphal aphid development time exhibited little variation across their shared interface, and the nymphal development time showed no changes in terms of mutual interactions.
Question 17: Figure 3 and 4: M. persicae without parenthesis. For what purpose the marks a, b, c were placed in the caption if they do not exist on the chart? What is BSA meaning?
Answer: Both the figure captions have been modified
Figure 3. Mean developmental time, (±SE) of different nymphal instars of M. persicae on chili plants by PeaT1 elicitor protein at different concentrations and different temperature (23, 25, 27°C) regimes (n = 10). Different alphabets above bar tops specify significant differences amongst treatments (factorial analysis one way ANOVA; LSD at α = 0.05).
Figure 4. Mean developmental time, (±SE) of different nymphal instars of M. persicae on chili plants by PeBC1 elicitor protein at different concentrations and different temperatures (23, 25, 27°C). regimes (n = 10). Different alphabets above bar tops specify significant differences amongst treatments (factorial analysis one way ANOVA; LSD at α = 0.05).
Question 18: Figure 5: Do not put “PeaT1” in the axis name; remove “with (a, b) ± SE” from the caption; check the correctness of the letters above bars at 27°C
Answer: The figures have been modified as per your kind suggestions in the revised version of the manuscript
Question 19: L 522-524: The sentence is confusing. Did the authors wish to express that the aphid fecundity was significantly affected by different concentrations of PeaT1 and PeBC1 and temperature regimes?
Answer: Yes, the sentence has been modified more undoubtedly in a revised version of the manuscript i.e. aphid fecundity was significantly affected by different concentrations of PeaT1 and PeBC1 and temperature regimes, as shown in Table D1 and D2 of Appendix D.
Question 20: L 527-530 “However, a minimum fecundity at a maximum temperature was observed at 27° C, and a maximum fecundity at a minimum temperature of 23 °C was recorded.” In my opinion, it is true only for seedlings treated by elicitors (Figure 5 and 6).
Answer: Yes, the sentence structure has been modified more clearly in a revised version of the manuscript i.e.
However, a minimum fecundity of aphid at a maximum temperature was observed at 27° C in seedlings treated with elicitor proteins-PeaT1 and PeBC1, and a maximum fecundity of aphid at a minimum temperature of 23 °C was recorded in seedlings treated with elicitor proteins-PeaT1 and PeBC1.
Question 21: L 37-39; 548-550: There are no data about trichomes and wax structure on seedlings (any photo. table, figure?). The finding “more trichomes” is not a research result.
Answer: The authors are extremely sorry about this error made due to the large amount of data set in the manuscript unfortunately we missed this important data now it has been incorporated in the revised version of the manuscript i.e.
The surface of chili leaves was significantly modified by PeaT1 and PeBC1 elicitor proteins, and seedlings treated with PeaT1 (56.23 ± 0.42 mm-2) and PeBC1 (52.14 ± 0.34 mm-2) exhibited more trichomes compared to control (30.17 ± 0.16 mm-2).
Question 22: Figure 7: What is PeBL1 in the caption?
Answer: By typing mistake, it has been replaced in the revised version of the manuscript
Question 23: Figure 8: Check the letters above bars please, are they put correctly?
Answer: all letters have been cross-checked and revised accordingly in a revised version of the manuscript
Question 24: Figure 9 and 10: word sequences are repeated
Answer: It has been changed and written separately for both elicitor proteins in a revised version of the manuscript
Question 25: L 660-663: The sentence is hard to follow.
Answer: The sentence structure has been modified in a revised version of the manuscript
PeaT1 and PeBC1 reduced disease severity by activating a photo-synthetic process that mimicked plant development features, as well as improving induced resistance in PeaT1 and PeBC1-treated chilli seedlings [66][67].
Compared to the controls, the PeaT1- and PeBC1-treated seedlings and leaves possessed more trichomes
Question 26: L 683: Plants couldn’t produce aphids! Aphids produce fewer offspring on PeaT1 and PeBC1 treated plants/seedlings as compared to water and buffer treated ones.
Answer: The sentence has been revised in a more clear way i.e.
Aphids produce fewer offspring on PeaT1, PeBC1 treated plants/seedlings as compared to water, and buffer treated ones.
Question 27: L 690: “on” plants not “in” plants
Answer: it has been modified in a revised version of the manuscript
Question 28: L 687-694: The effect of temperature on aphid development is well known. The authors probably wanted to demonstrate the inhibitory effect of the studied elicitors on the development and fertility of aphids regardless of temperature. Thus the sentence should be reconstructed.
Answer: The recommended suggestions have been followed and highlighted in the revised version of the manuscript i.e.
Additional studies need to be conducted to understand the underlying mechanism of PeaT1 and PeBC1 on chili plants, particularly the inhibitory effect of both elicitors on nymphal development time and fecundity.
Question 29: L 755: EPG study doesn’t confirm resistance due to strict (e.g. trichomes) physical features or chemical composition of the plant. It only shows the aphid feeding mode. It is not possible to say unequivocally what factor caused the feeding difficulties.
Answer: Statements about EPG have been revised i.e.
Likewise, an EPG study presented and gave an idea that the resistance induced by PeaT1 and PeBC1 might be due to the modification of physical defense and an increased number of trichomes, while the wax composition and antibiotic or antixenosis effects affected aphid feeding behaviour in PeaT1- and PeBC1-treated seedlings

Reviewer 2 Report
Microorganisms MS #-1421498
I prevrously reviewed the original version manuscript. The author did a great job to address my comments. I see that this version is highly improved. The introduction provided enough background about the topic. The discussion section is highly fixed and well written. I have minor conserne about the title and the abstract.
1-Title is too long
2-Also the abstract , Please add more values that represent the most significant data
Other than that, this manuscript is scientifically sound
Author Response
Reviewer 2 Comments
Question 1: I previously reviewed the original version manuscript. The author did a great job to address my comments. I see that this version is highly improved. The introduction provided enough background about the topic. The discussion section is highly fixed and well written. I have a minor concern about the title and the abstract. Other than that, this manuscript is scientifically sound, 1-Title is too long Please add more values that represent the most significant data?
Answer: Dear reviewer the authors are thankful for your patience and guidance towards this manuscript, the title has been revised further as per your valuable suggestions, also the abstract the values data set have been added in the abstract as well in a revised version of the manuscript
This manuscript is a resubmission of an earlier submission. The following is a list of the peer review reports and author responses from that submission.
Round 1
Reviewer 1 Report
Please see the attached word file for comments and suggestion

Reviewer 2 Report
Review – MS ID: microorganisms-1382407
Title: Potential use of Electrical Penetration Graph (EPG) to induce resistance in Chilli (Capsicum annum) plant against Green peach Aphid (Myzus persicae) by Microbial Protein Elicitors PeBC1 obtained from Botrytis cinerea and PeaT1 from Alternaria tenuissima
- The Authors conducted several experiments and obtained a large number of results, which, however, are analyzed and presented superficially.
- The manuscript is written carelessly. The subject is interesting, but the language of the manuscript need to be improved by a native speaker familiar to the subject. The manuscript contains a large number of misleading statements, stylistic errors, typos, etc. in the current version. In my opinion manuscript should be rewritten, because scientific quality of all paragraphs is very low.
- The Title and Abstract are inappropriate to the content of the manuscript.
- The Introduction and Discussion sections are sketchy and should be reinterpreted.
- No research hypothesis nor novelty of the research were presented.
- The illustrative part also requires improvement, e.g. Table 1, Table 2, Figure 4
In conclusion, present status of manuscript has not reached to the considerable scientific quality to be published in Microrganisms (a Q2 journal).
Reviewer 3 Report
General comments:
-In general, this manuscript has a valuable topic. The topic is scientifically sounds.
- The manuscript is well written except for minor English language check required.
-The experimental design is adequately discussed.
- My main concern was the introduction and the discussion section.
-There are some minor comments.
Detailed comments:
Title:
The title is too long. Please change the title to be shorter and more précised.
Abstract:
This section is missing the direct aim of the study, please write the aim as following; The aim of this study was….
-Please include more values and numeric numbers for the most significant findings
Keywords:
Please add EPG to the keywords list
Introduction:
This section doesn’t provide enough background about the topic. I see that the introduction is very poor. This section needs to be enriched and provided with more background about the topic.
Materials and Methods:
The experimental design was suitable and adequate to the current study.
Results:
The results were well presented.
Discussion:
I had a hard time to make the connection between your discussions and the results in which figure.
Please make subheadings in this section in organized way, the same order as the subheadings in the results section.
-Please rewrite this section in more organized way and relate to the data in tables and figures carefully with a comparison to the previous studies.
If it is easier to discuss the results clearly and thoroughly, the author is advised to combine the results and the discussion in one section for the best presentation and discussion to the results.
Conclusion:
This section was well written and provided a good conclusion for the study and include the significant findings.
References:
The authors provided enough citations, and it was UpToDate.
*I am convinced that this manuscript is very valuable and will be suitable to be published in microorganisms journal after minor revision.